# Fake news, misinformation, vaccine hesitancy and the role of community engagement in COVID-19 vaccine acceptance in Southern Ghana

**Mawulom Kuatewo[1], Wisdom Ebelin 🔾[2]\*, Phidelia Theresa Doegah[3], Matilda Aberese-Ako[3], Samuel Lissah[4], Atsu Godsway Kpordorlor[3], Lebene Kpodo[5], Senanu Djokoto[6], Evelyn Ansah[3]**

**1** Hohoe Municipal Health Directorate, Ghana Health Service, Hohoe, Ghana, **2** Evangelical Presbyterian Health Services, Evangelical Presbyterian Headquarters, Ho, Ghana, **3** Institute of Health Research, University of Health and Allied Sciences, Ho, Ghana, **4** Department of Agricultural Sciences and Technology, Faculty of Applied Sciences and Technology, Ho Technical University, Ghana, **5** Savana Signatures, Tamale, Ghana, **6** Regional Health Directorate, Ho, Ghana

\* ebelinwisdom@gmail.com

## Abstract

### Introduction

The novel coronavirus (COVID-19) is characterised by loads of fake news and misinformation, which can influence vaccine acceptance. Implementing a harmonized public health strategy during an outbreak necessitates effective community engagement and communication, which facilitates public trust and decision-making. This study explored the role of community engagement in the acceptance of COVID-19 vaccine amid fake news and misinformation in two municipalities in Ghana.

### Method

A case study design was employed using in-depth interviews with government officials from the Ghana Health Service, Municipal Assembly, Information Services Department and the National Commission on Civic Education and community gatekeepers. Additionally, focus group discussions were conducted with a cross-section of women, men and migrants' community members to understand the role of community engagement in vaccine acceptance. Qualitative analysis software Nvivo 12 was used to support thematic coding and analysis. All ethical procedures and COVID-19 preventive protocols were observed.

### Results

Study participants reported the sources of fake news and misinformation about the COVID-19 vaccines from interpersonal communication, the radio, and a popular anti-vaccine song. Some of the factors contributing to vaccine hesitancy were

**Data availability statement:** Data cannot be shared publicly because of participants confidentiality. Data access requests may be made to the Research Ethics Committee at the University of Health and Allied Sciences, Ho, Ghana. Address: PMB 31, Volta Region, Ghana Email: rec@uhas.edu.gh Telephone: +233 362196193

**Funding:** Initials of the authors who received each award: MA Grant numbers awarded to each author: Royal Society of Tropical Medicine and Hygiene Early Career grants programme The full name of each funder: Royal Society of Tropical Medicine and Hygiene URL of each funder website: https://www.rstmh.org/rstmh-early-career-grants-programme-%E2%80%93-guidance-2024

**Competing interests:** The authors have declared that no competing interests exist

community members believed in the fake news and misinformation, low trust in the government and public institutions, and the lack of extensive education on COVID-19 vaccines. The Ghana Health Service was the most successful in engaging communities to promote vaccine acceptance amid fake news and misinformation. It leveraged on its existing community-based health planning and services (CHPS) programme, which engaged the communities frequently through routine programmes such as durbars, antenatal clinics, child welfare clinics, and other community programmes to carry out engagement.

## Conclusion

Misinformation and fake news about COVID-19 vaccines were widespread in the study communities, with significant implications for vaccine hesitancy. The sources of misinformation ranged from social media platforms and radio broadcasts to personal interactions within communities. While government efforts at community engagement were noted, these efforts were often inadequate to counteract the deeply ingrained fears and misconceptions.

---

## 1. Introduction

Pandemics claim millions of lives when they strike [1]. In the absence of an effective treatment, the right information effectively communicated can save lives [2]. This therefore requires that health information is provided by scientific experts, medical personnel or health authorities [3]. Effective communication builds public trust and encourages people to adopt a recommended behavior [4]. In the case of the novel COVID-19 pandemic, vaccination has been recommended as the most effective way of curbing the spread of the disease and saving lives [5]. Sadly though, epidemics according to [6] are often characterized by loads of fake news. For instance it has been reported that during the Ebola epidemic, there was the fake news that anti-Ebola tablets had fatal consequences on people and that the vaccine was effective only for people from the white race [6]. Similarly, during an outbreak of Hepatitis A in San Diego between 2016–2018, there was dangerous misinformation that the vaccine could lead to autism. The novel coronavirus (COVID-19) has been characterised by huge amounts of misleading and false information about the virus and the vaccines [7]. According to van Der Linden et al. [8], information that is inconsistent with the facts about the natural history, epidemiology, and clinical outcomes of COVID-19 have been produced from all kinds of sources, including the political and scientific world as well as from the social media [9]. Bernard et al. [9] have noted that false stories about COVID-19 have become so common that it has become difficult to distinguish fake news from the truth The World Health Organization (WHO) has in fact warned the public that the world is in battle over another type of epidemic called 'infodemics' which it defines as the spread of fake news, false information, and false scientific claims. In Vietnam for example, evidence shows that fake information on COVID-19 is more than the official and reliable information given by the authorities [10]. Similarly,

studies show that fake news and misleading information have become so popular in China and India that the Indian government has resorted to the drastic measure of turning off the internet to curb the spread of rumors [11]. In the U.S.A, extensive use of social media caused many people to share and rely on false information related to the corona virus [12].

Sub-Saharan Africa has been flooded by tons of fake information related to COVID-19 [14]. According to Osuagwu and others, one such misinformation is that COVID-19 has only slight effects on Blacks compared to Whites [13]. In Nigeria for example misinformation and fake news proliferated mainly because the sharers of those news thought they were helping others just as their culture demanded [10]. In Ghana, limited community education fuelled proliferation of misinformation on COVID-19 and its vaccines such as the COVID-19 vaccine is a chip being implanted in the human body and have resulted in fear, vaccine hesitancy [14], and has been observed as a threat to public health [10]. According to WHO, culturally appropriate communication can give positive results during an outbreak [15]. Additionally, implementing a harmonized public health strategy during an outbreak necessitates effective communication [16] because effective communication is the main public trust driver [17]. Research evidence has also shown that people are more likely to trust government and believe in vaccination only if they are properly and effectively educated [18]. Not only does communication clear doubts in the public's mind but also helps them make the right decision as they will neither over-nor underestimate the severity of the outbreak [19].

According to the WHO, vaccine hesitancy is a severe threat to global health [20]. Vaccine hesitancy is defined as a delay in the uptake or the refusal of vaccines, despite the availability of vaccination services [21]. Several factors can be posited for vaccine hesitancy such as religious persuasion, misinformation, and lack of proper orientation regarding the scientific basis of vaccines and the result of widespread misinformation across multiple social media channels [22]. The Ghana Government made several efforts in engaging its citizens to accept the COVID-19 vaccine through some governmental institutions. The Ghana Health Service (GHS) regularly held press briefings to provide a situational report on COVID-19 and educate the populace about the COVID-19 vaccine. Also, health personnel involved in the vaccination program sensitized the general public about the safety, efficacy, possible side effects, and benefits of receiving the COVID-19 vaccine, which has aided in the acceptance and uptake of the vaccines [23]. Furthermore, to promote uptake of the vaccine, the Ghana government adopted multi-faceted health education approaches, such as use of print, electronic and social media. These communications strategies were led by the Ministries of Information and Communication, Information Services Department, Ministry of Health and Ghana Health Service (GHS) [24]. Ghana administered 28,515,854 doses of the COVID-19 vaccine as of December 2023 and as of April 07, 2024, the country had 21 active cases [25]. The country dedicated the month of July 2024 to intensify efforts at administering 500,000 doses of the COVID-19 vaccines to persons 18 years and above [25]. Despite government efforts to promote the uptake of the COVID-19 vaccine, findings of a study conducted by Manu et al [26], indicated that misconceptions about the COVID-19 vaccine persists in the Volta Region. This qualitative research study therefore explored how fake news and misinformation influenced vaccine hesitancy and the role of community engagement in promoting vaccine acceptance in Volta Region, Ghana.

## 2. Methods

This section presents details of the methods used for the study.

### 2.1 Study design

A case study approach was used, which employed in-depth interviews (IDIs) and focus group discussions (FGDs) to explore and describe how community engagement was utilised to promote COVID-19 vaccine acceptance in the midst of fake news and misinformation in two municipalities in the Volta Region of Ghana.

### 2.2 Selection of study areas

The Volta Region was chosen for the study because it recorded a low vaccine acceptance rate (32.50%) [26]. Two municipalities dubbed Municipality A* and Municipality B* were purposively selected for the study, because they were among

the most urbanized municipalities in the region. The choice was aimed at understanding how urban populations engage with the healthcare system and other governmental entities in providing health interventions. Both municipalities have a fair proportion of the rural population and Municipality B also has migrant settlements mostly made up of persons from the Republic of Togo, with which it shares boundaries.

The Ghana Health Service operates a decentralized administrative system with offices at the national, regional, municipal levels, sub-district offices, hospitals, health centers, and Community-based Health Planning and Services (CHPS) compound in each region [27]. To ensure that the study was reflective of the different levels of health service delivery, a multi-stage random sampling technique was used to select a sub-municipality and a CHPS facility for the study. In the first stage, the names of all the sub-municipal health directorates in each study municipality were obtained from the respective Municipal Health Directorates. They were written on pieces of paper, which were folded and an observer selected one sub-municipality each from the two municipalities for the study. In the second stage, all CHPS compounds under the two selected sub-municipalities were written on pieces of paper, which were folded and the observer randomly picked one. To ensure that community experiences were also captured in the study, a third stage was included, which concerned writing down the names of communities under each selected CHPS zone, folding it, and letting the observer randomly selected two (one from each municipality) to participate in the study.

## 2.3  Selection of study participants and sampling

From each of the study sites, one Municipal Health Directorate Officer, one sub-municipal health leader, two health providers from the selected CHPS facility, one official from the Municipal Assembly, NCCE, and ISD respectively, and some of the community elders were purposively sampled to participate in IDIs (Table 1). A cross-section of women, men, and migrants of different age groups who were available and willing to participate in the study were conveniently sampled to participate in FGDs consisting of 6–13 participants. The interviews sought to understand how the Municipal Assembly, GHS, NCCE, and the ISD engaged communities toward the COVID-19 vaccination rollout and uptake. Saturation was attained when no new information was obtained from study participants, which is in accordance with qualitative inquiry [28]. Participation was voluntary and those who were not interested were automatically excluded, as well as those who were mentally challenged.

The three components of the WHO's definition of community engagement (categories of stakeholders, processes used to engage communities, and the purpose of the engagement), were used to guide the selection of the stakeholders, design of the IDI and FGD guides and to determine the focus of the study, which was on COVID-19 interventions (details of the questions for each spectrum of engagement has been included as appendix 1). This manuscript is drafted from a larger study and other aspects have been published [29].

## 2.4  Training of data collectors and quality control

Four graduate data collectors were trained by the fourth author, MA, a medical and organizational anthropologist, on community entry, qualitative data collection methods including interviewing, writing field notes and seeking informed consent, which equipped them with the needed skills for data collection. The IDI and FGD guides for community members were translated into the native language Ewe during the training and research assistants were trained in English and the local language.

The study guides were pre-tested among eligible participants from a municipality similar to the municipalities selected for the study. The pre-testing was done to assess the clarity, relevance, and appropriateness of the questions to elicit the intended responses that align with the study objectives.

## 2.5  Data collection procedure

Using in-depth interviews (IDIs) and focus group discussions (FGDs), the study explored in-depth knowledge on the Ghanaian government's initiative through government institutions such as the Municipal Assembly, Ghana Health Service

**Table 1. List of data collection methods and categories of respondents.**

| Study participants who participated in IDIs | Municipality A | Municipality B |
|---|---|---|
| District Health Officials (1 municipal and 1 sub district) | 2 | 2 |
| Frontline workers in CHPS compounds | 2 | 2 |
| District Assembly officials | 1 | 1 |
| National commission for civic education | 1 | 1 |
| Information services department | 1 | 1 |
| Chiefs and queen mothers | 2 | 2 |
| Community Elders | 2 | 2 |
| Religious Leaders (Christian, Moslem, Traditionalist | | |
| Herbalists | 2 | 2 |
| Assembly persons | 1 | 1 |
| Community healthcare volunteers | 2 | 2 |
| **Total** | **18** | **18** |
| **Study participants for FGDs** | | |
| Women below 30 years | 8 | 8 |
| Women above 30 years | 10 | 8 |
| Men below 30 years | 9 | 9 |
| Men above 30 years | 9 | 8 |
| *Migrant men | 0 | 7 |
| Migrant women | 0 | 7 |
| **Total** | **36** | **47** |

*Municipality A does not have migrant settler communities; thus, no interviews were conducted for such a category

Source: Field work 2021

(GHS), Information Services Department (ISD) the National Commission on Civic Education (NCCE) and community experiences. Data were collected from May 16, 2021, to August 27, 2021. The research team comprised of a female medical anthropologist (MA) and 4 graduate research assistants (RAs). Two of the RAs were females and two were males, who could also speak the indigenous language of their assigned study areas.

In-person interviews were conducted in English with the government officials (health workers, NCCE officials, ISD officials and assemblymen) and in Ewe with the community members. Migrants were interviewed in French, as majority of them were from Togo and could not speak English nor the Ewe dialect spoken in Ghana. Interviews were recorded using a digital audio recorder and later transcribed verbatim to preserve respondents' views and experiences. The average duration of IDIs was 50 minutes and FGDs was one hour. Meetings were held between MA and the data collectors every week to ensure the trustworthiness of data.

### 2.6 Data management and analysis

Transcribed data (IDIs and FGDs) were uploaded onto a computer and transferred onto a qualitative software NVivo 12, to support data coding. The data was triangulated and analyzed thematically. Deductive and inductive coding were carried out by LK, WE, AK and MK through carefully reading data, thinking critically and paying attention to the study questions, which were based on the five spectrum of community engagement [30]. MA validated the codes by cross checking them with the study questions and with a sample of study participants' responses. Matrixes were developed from the coded data to support further analysis. The themes were generated from the analysis report on how government institutions conducted engagement activities in the two study municipalities to address vaccine acceptance amid fake news and misinformation. Other aspects of the study have been reported in another article [31].

### 2.7 Maintaining rigour

Rigor was maintained through employing triangulation in data collection by employing a mix of IDIs and FGDs, and during the analysis process. Also, at the end of each day of data collection, the principal investigator and the data collectors held a meeting to debrief and to reflect over the study to guide towards meeting the objectives of the study. Credibility and reliability of the study's findings were carried out by organising dissemination in each of the study sites where the results were presented to the study participants for them to confirm the accuracy of the results.

### 2.8 Ethics

The University of Health and Allied Sciences (UHAS) Research Ethics Committee (REC) approved the study (UHAS-REC A.5 [5l 20−21]). Potential study participants were approached and informed of the study and those who were willing to participate were taken through a consenting process. The consent form contained the following information: purpose, procedure, contact details of the principal investigator and the REC administrator, plan to disseminate study findings, date of consent, and signature columns for the interviewer and interviewee. Two copies of the consent form were completed, one was given to the study participant, and the second copy was kept by the study team. Participation was voluntary and those who were not interested were automatically excluded, as well as those who were mentally challenged and persons under 18 years old. COVID-19 protocols were observed throughout the study. Interview participants were offered disposable masks, and their hands were sanitized before consenting and participating in the interview. The data sets were anonymized to protect study communities and participants' identities and were accessible to only the study team.

Community entry was carried out in the study municipalities and study communities. The study team visited and sought permission from all gatekeepers (municipal chief executives, municipal health directors, CHPS compounds, officials of the NCE, and ISD, Assembly members, chiefs, and opinion leaders in the two municipalities).

## 3. Results

Findings are presented on the types and sources of misinformation and fake news concerning the COVID-19 vaccines and factors that contributed to their proliferation in the study communities. Results on how government institutions dealt with the misinformation and the effect their response had on the community members' willingness to vaccinate were also presented.

### 3.1 Fake news and misinformation

All study participants reported that they were aware of the rumours about the vaccines. These rumours according to most of the participants were negative, while a few stated that they heard positive stories. Study participants who reported that they heard negative information about the vaccines said they heard the vaccine was aimed at reducing the population of Africa. They explained that the vaccine was aimed at making African men impotent and African women infertile. Majority of the participants also reported that the vaccine could cause disability or death, either instantly or a few years after vaccination. Study participants also claimed that they heard that the COVID-19 vaccines administered in European countries were different from the ones being administered in African countries including Ghana. A few of the participants reported that they heard that the vaccine could result in Africans getting inclined towards homosexuality, while others reported that the vaccine could render Africans mentally retarded so that the whites would be able to take over the resources in the future. Others reported that they heard that the vaccine was a form of a sim card being injected into people's bodies, to make it possible for whites to control them remotely and a participant reported that he was told that those who took the vaccine turned into animals.

> "*When we hear these stories, we become scared. When they developed the vaccine in the US, they halted its use but they are telling us to take it. There is no company in Africa that produces vaccines. It is only those from the outside world and these guys have plotted to make us extinct. We are scared of the vaccine. The whites want to take over*

*Africa due to our natural resources, so they want to decrease our population. They are giving us this vaccine to make us infertile or to cause us to die. They are experiencing tsunamis and other natural disasters, so they want to run to Africa.... I heard that when you take the vaccine, after some years to come, a man could meet his fellow man on the street and start having sexual feelings for him. Due to the vaccine men could sleep with their fellow men. These whites hid something in the vaccine which can be used to control you from wherever they are located. You will only do their biddings unknowingly or unwillingly.*" (FGD Participant, Male Migrants, Municipality B)

"*The concern that people shared was that the whites that developed the vaccine also at times make some medicines that are poisons. The doctors do not know it is poison, they see it only as vaccine. The same way, the coronavirus came and everyone is afraid of its vaccine, saying that if they should take it, their life expectancy will be reduced. That is why people do not want to take the vaccine.*" (Male volunteer, Municipality A)

On the other hand, a few participants were also misinformed that the vaccine could prevent COVID-19 completely and give long life to those accepting it and that it could prevent any other disease: "*They said that when we take the vaccine, no disease will affect us.*" (FGD Females less than 30 years, Municipality A)

Study participants reported that fake news and misinformation made them afraid and discouraged them from getting vaccinated.

"*Our president his Excellency Nana Akufo Addo brought a number of them [COVID-19 vaccines] and they injected the health workers earlier. Recently I heard that he said he has made plans for 20,000 vaccine doses for Ghana. But the fear …is that there is no assurance about the safety of the vaccine and whether it will be beneficial. So, I myself have said that when the vaccine comes, I will not take the injection because of the things and the information I am hearing.*" (Pastor- Municipality A)

### 3.2 Sources of misinformation and fake news

Majority of the study participants reported that they heard fake news about the COVID-19 vaccine from friends and relatives in their communities and also from the media such as the radio and television. A few participants also indicated that they heard such news from acquaintances who live abroad and through newspapers.

"*But COVID-19 vaccination, they said AstraZeneca vaccination, we heard it in the news, on radios, in papers, media and internet that Indian government is the one producing this vaccine.*" (Pastor- Municipality A)

"*I forgot some important points… These whites hid something in the vaccine which can be used to control you from wherever they are located. You will only do their biddings unknowingly or unwillingly. The most intelligent Africans who are all abroad are feeding us with some of these information.*" (Male participant, FGD, Migrants Municipality B)

Interestingly, all the study participants also reported having heard that the vaccine was not good by listening to a song composed by a Ghanaian artist in one of the Ghanaian languages (Ewe). Another popular source was through social media platforms particularly WhatsApp platforms.

"*No matter how ill I am, I only use herbs. When I squeeze the herbs with water and drop some lemon in it. That's all I use. Whatever illness comes I only use herbs. I say even if I'm coughing, I will only chew ginger. That injection I am not taking it. The injection I'm not taking it. That injection I'm not taking it. The thieves have deceived themselves. The injection I'm not taking it…If I get a Dzemeni herb [explain the Dzemeni herb here] and mix it with neem tree leaves and mix it with 'devetsui' (local name of a certain herb) and add some milk, then I've become a pharmacist. …My body is very strong.*

*That injection I'm not taking it. Never…My mother won't take it; my wife won't, my child won't. …I will follow the ways of my forefathers. All their teachings I no fit forget. Full Ewe man [indigenes from the Volta region], I won't fall in any trap that the white is setting. I will only stick to the herbs. It has always been there for us since ages and I came to meet it. The injection I'm not taking it. I'm not taking it."* (Lyrics of the song titled Abi, translated from Ewe into English by authors, 2021)

*"Also, it has spread on all social media platforms that the vaccine is fake, and some people also did videos and voice notes on WhatsApp platforms that they should not accept the vaccine. So that is what destroyed everything. Many people heard it because everybody got up and started sharing it on social media platforms. So that information reached the entire community and people do not trust anything about it again. It will be very difficult before it will completely go off the minds of people."* (Community Elder, Municipality A)

*"It is because of what we heard that they will come and inject us to die, they want to clear off human race, they want to clear off the blacks and all that. When corona came, there were a lot of rumours on our WhatsApp platforms."* (Community Elder Municipality B)

### 3.3 Factors accounting for belief in fake news and misconceptions and the willingness to vaccinate

Interactions with study participants suggested that factors such as lack of trust in the Ghanaian government contributed to the spread and success of fake news.

Study participants distrusted the Ghanaian government because of the varying number of COVID-19 cases reported by the Ghana health service. Sometimes the cases rose and other times it dropped. They felt that the government's motive was to mislead the general populace to raise funds for some individuals to benefit. Also, study participants believed that the government was corrupt, so the West was using it by giving it money in order to get the government to mislead the citizenry to take the vaccines. Consequently, they were unwilling to vaccinate.

*"…the cases, as they say it has increased in Ghana, it is up, it is down, it is a camouflage from the Government to raise funds into the pockets of some individuals to use…So, that trust was not there for some time. … we do not have 100% trust in the government concerning COVID-19."* (Church Leader, Municipality A)

*"Our governments are like idols. When their faces are turned in a direction because of money; that is the only thing they see. These whites know they can only get us through the government. They gave them money to entice us so that when we take it, they give us something small and they get to keep the rest of the money. The government cannot tell me to take the vaccine and I will go for it, never. It is a deception."* (FGD, Male Migrants, Municipality B)

Some of the study participants believed that the vaccine was effective, however, the health workers were not competent, so they were administering the vaccine wrongly. The level of distrust was extended to childhood immunisations as community members became suspicious of health workers who usually visited their homes to vaccinate children, which resulted in the refusals of such vaccines.

*"Before any drug will be imported into the country, it is checked with the computer to ascertain its potency and efficiency. So, I believe the vaccine is good, but the healthcare workers didn't give the injections well and that is what led to the problems."* (FGD, Male Above 30 Years, Municipality B)

*"… Even it came to a point where they [health workers] cannot even go to the houses to give normal [routine childhood] immunizations. People will not even bring out their children to be immunised, because of the corona [COVID-19] vaccine. They are afraid that it is the coronavirus vaccine"* (Community Elder, Municipality B)

Study participants mostly relied on other community members, rumours, and radio programs as their main source of information on the COVID-19 vaccine.

*"There are a lot of misconceptions about this vaccine, fertility issues, so many. They talk about it anytime we go for communal labour: 'Oh the government wants to kill us; they want to stop us from giving birth.'. So, a lot of them do not really get information about the vaccine... sometimes when we hear it, we just have to come in and give them education."* (Assemblyman, Municipality A).

*"I've not been to school before. I am an Okada rider. I only work with the hearsays I*

*hear around."* (FGD, Migrants Male 30 years and above, Municipality B).

Many radio stations and TV stations reported the fake news that were circulating on the COVID-19 vaccine. Most participants demonstrated a high trust in the media, so once any information was reported by the media, they believed it.

*"Yes, we believe everything that comes from these media platforms, they are so valid. So especially when you hear it on radio. Like these elderly ones, they will say 'oh, we have all heard it on radio already.'"* (Church Elder, Municipality B)

*"On the radio station. I heard on Citi that there is no assurance about the safety of the vaccine and whether it will be beneficial. I said that I heard it on Citi Breakfast news from Mohammed\* [a Ghanaian journalist] and his team. I heard it from Lorlornyo major news as well as Heritage FM. GBC too. All the media houses all over. I heard it personally."* (Church Leader, Municipality A)

Some participants believed that COVID-19 was not yet in Ghana, while others were of the view that the number of cases reported in the country was less than what was being reported by government officials. Others said that the burden of the disease had reduced in the population, so, there was no need for a vaccine, and easily believed or spread the fake news they heard about it.

*"… the disease has not come to Africa, so why are they bringing the vaccine!"* (Community Elder, Municipality A)

*"Even if the government brings the vaccine right now, it would be difficult to take it, because as it stands now, the burden of the disease has reduced, and it is left with us to take care of ourselves and follow the safety protocols."* (FGD, Females 30 Years and Above, Municipality B)

Some study participants who experienced severe adverse events following Immunization with other antigens in the past, whilst others also witnessed close relatives experience adverse side events after taking the COVID-19 vaccine were reluctant to vaccinate. As such they found it easy to believe and inform others that the vaccine was harmful.

*"…, when one of my children took the vaccine, it really disturbed him. The injection site got swollen and he had to go to a different hospital to be attended to before he was fine. One of my siblings also had a swollen face after taking the vaccine; so it has created some fear in me."* (FGD, Female Participant above 30 Years, Municipality B)

Another factor for the refusal to vaccinate had to do with individuals' personal perceptions on vaccination. Some reported that they were afraid of needles and being injected with them.

*"Now, in fact, some of us do not like taking injections. Not only the COVID-19 vaccine. Like coming to the hospital right now and you tell me that you are going to inject me, then you will not see me again... So, I fear a lot that they will use a needle and pierce me."* (Church Elder, Municipality B)

Few study participants had a misconception that the vaccine was meant to treat COVID-19, so, there was no reason why they should vaccinate since they were not sick. Others on the other hand, still sought to know whether the fake news and misconceptions were true.

*"Those who test drugs before we accept them into our countries should not accept any bribe so that even if the vaccine is not good, they will be administering it to us? And here lies the case, they said after vaccination, I need to wear the nose mask, still sanitize, and practice social distancing. So why should I vaccinate? I have been following all these protocols and have not contracted the disease. Why then should I get vaccinated and experience the reactions it comes with?"* (Opinion leader, Municipality A)

*"I just want to know if what (referring to the rumours they heard about the vaccine) we heard about the vaccine are true,"* (FGD, Migrants Females Municipality B)

Some of the study participants reported that they had not received adequate information on the COVID-19 vaccines and so they were not willing to take it. Others reported that they were not properly informed by the government on the Covid-19 vaccine. As such, they indicated that they had not received ample education about the vaccine and so they needed to be educated in order to help them get over the misconceptions before they will take the vaccines.

*"I will think about it because we have not received in-depth teaching about the vaccine and due to that, we have kept the notion that they want to reduce our population. If the vaccine is brought now, I would not take it. However, when there is in-depth teaching and I understand and accept the teaching, I would take it."* (FGD, males 30 years and above, Municipality B)

A noteworthy factor that made it easy for misinformation on the COVID-19 vaccine to thrive was the community members previous experience with the truncated Ebola trial in Ghana. The study region was earmarked as an Ebola trial centre. However, Participants believed that they were being used for a clinical trial on the COVID-19 vaccine.

*"For the COVID-19 vaccine, I have not heard anything about it but what I heard from the time of Ebola was, normally in the Volta Region, Municipality A and certain areas of Oti are used for vaccine testing. It came up during the Ebola pandemic."* (Opinion Leader, Municipality A)

Study participants also reported that they had alternative sources of care such as herbs and alcohol that was working for them. They believed their herbs were powerful enough to offer them immunity against COVID-19 and even cure them if they got COVID-19. Some added that their forefathers took herbs and lived longer, so they too are taking herbs and that is what is sustaining them

*"Taking herbal medicine is better. It is the herbal medicine that our forefathers too drank, and they lived longer. I have also been using the herbs and got to this stage. So, if you are taking a good herb, you will live long."* (IDI, Herbalist/Fetish Priest, Municipality B)

*"My colleague just mentioned the strong alcohol. However, when you get herbs that have not been polluted by fertilizers, they would get you healed. This is because even when you go to the hospital, the drugs are made from herbal extracts."* (FGD, males above 30, Municipality B)

*"Our herbs are more powerful than that vaccine."* (FGD, females above 30 years, Municipality B)

### 3.4 Government's community engagement efforts to counter fake news and misinformation and its influence on communityacceptance of vaccination

The Ghanaian government through its institutions such as the municipal assembly, the Ghana Health Service, the Information Services Department, and the National Commission for Civic Education carried out various community engagement activities such as meeting with community groups, elders, and community members to inform them about the vaccine and to encourage them to take the vaccine.

The Ghanaian government's strategy to dispel the fake news concerning the COVID-19 vaccine included a weekly broadcast of the president's COVID-19 update, the televising of the president, the vice president, their spouses, and other prominent dignitaries such as chiefs and religious leaders receiving their COVID-19 vaccine. Health workers and other essential workers in the fight against COVID-19 were also among the first group of people to take the vaccine. Some of the government officials such as health workers, NCCE officials, and ISD officials also took their vaccines in public, while some of the health workers recorded them and showed the videos and pictures to their clients and communities within their districts to motivate community members to vaccinate against the COVID-19 virus. *"It is through education and they was the president taking the vaccine, the vice president, the former president Kufuor on the television* (Information Services Department Director Municipal B).

Government officials reported that they engaged the various communities through existing mediums such as community durbars, opinion leaders, religious bodies, various associations such as drivers and traders' associations, and radio programs.

*"Initially we had this public health emergency management committee. So as part of their plans, they identify various groups in the community. For example, we have the market women or market traders' groups, we have drivers' unions or moto bike riders and other associations of small-scale industries operating in the municipality, then we also have the chiefs. So, we engage these groups at a particular time. When we engage them, we also listen to them, and we give them the message on how to prevent the spread of the disease. We also encourage the various institutions at the community level to engage the subjects and educate them on the disease. So, it was good. I can give you an example. For example, when we met the drivers, we told them to always make sure everybody is in a nose mask before getting in their vehicles. So if you're not in a nose mask, they insist that you buy one around and put it on before joining their vehicle."* (Deputy Director of Municipal A Assembly)

Also, majority of the government officials interviewed expressed concern that the President of Ghana and the Ghana Health Service had announced that the second dose of the COVID-19 vaccine would arrive four weeks after the first dose was administered. However, the second dose was delayed by over 12 weeks, significantly contributing to public distrust in the government and leading many Ghanaians to refuse to take the COVID-19 vaccine when it was made available.

*"We are already doing that. We go on radio every now and then to talk about the vaccine. But now that the people are demanding for the second dose and we are not getting it, we are not able to go to the radio station to talk about it. If we go and talk about it and they ask for the vaccine, what would we do? So, now, we are mute for the meantime. When we receive it, we would let them know that we have received it and I know they would come in their numbers to take it."* (Health Director of Municipal B)

*"With regards to COVID-19 vaccination, generally, my personal concern is the delay in the second jab. I think the delay is too much. When word goes out that the second dose can be taken after let's say a maximum of four weeks, it should be four weeks. Once it delays, it creates room for any misconceptions to come up. And those are what might be affecting the others who are willing to or considering taking the jab later from probably changing their mind. So, once we have given a timeline, let's stick to it."* (Sub District In-Charge of Health Center Municipal B)

The Ghana Health Service also utilised routine health care programs such as child welfare clinics, antenatal clinics, and post-natal clinics to inform clients about the vaccine rollout. They informed and educated the communities that some of the companies that manufacture the COVID-19 vaccines were the same multi-national companies based in the developed countries that manufacture vaccines for their children against poliomyelitis, yellow fever, measles, streptococcus pneumoniae and other childhood killer diseases; so if they wanted to wipe out Africans, they could have done so using earlier developed vaccines rather than the COVID-19 vaccines.

*"If you listen to the radio, you'd hear people saying they would not take the vaccine because it is a plan by the Western world to kill Africans and they'll never take it and they're urging the public never to get vaccinated. So, when the vaccine came, we used the radio stations to educate them and tell them vaccines have existed for long. We have been immunizing the children with polio, tetanus, and other vaccines."* (Director NCCE Municipal A)

### 3.5 Consequences of limited engagement in addressing misinformation on decision to vaccinate

The engagement process contributed to some of the study participants changing their mind towards vaccination, nevertheless, others had already been influenced by the misinformation and myths and for that matter were unwilling to be vaccinated.

A few participants were indecisive and testified that they might change their minds and take the vaccine if the following conditions are fulfilled, 1. They are informed and educated about the vaccine, 2. others take the vaccine, and nothing happens to them and 3. they are sure that they would experience no side effects after taking it.

*"When it is brought, I would wait for other people to take it and if nothing happens to them, I will also take it."* (FGD, female migrants, Municipality B)

*"That is what I have said in my own case. They have said that the government has gone for the vaccine to kill people. Per my thought, as I have earlier said, I believe the government cannot go for vaccines because its citizens are sick and then he will inject them and those who are not sick will also die. I do not believe that the government can do such a thing. But since then, we heard that the vaccine was received in some places and people received it. For me, I am also waiting for the vaccine so that when it comes, if some people will not take it or if I will not take it, then it is up to me,"* (Chief, Municipality A)

*"If I am well educated and I understand everything about it, I will take it. But if I ask questions and they are not well explained to me, then, I will not take."* (Opinion Leader, Municipality A)

Very few of the participants revealed they had no reservations about the COVID-19 vaccine while others reported that due to the fear that they have towards the COVID-19 virus, they will accept the vaccine.

*"I have no concerns. When it comes, I will take it."* (Community Elder, Municipality A)

*"The reason why I accept to take it is because of the fear of the virus. That is why I said when it comes, I will take it so that I won't contract the virus."* (Volunteer, Municipality A)

Despite the engagement efforts made by the government institutions to get community members to vaccinate some were not convinced, and some community members still distrusted the efforts from the president and other government institutions to vaccinate.

*"First, I do not know if the vaccine they will inject me with is the one they injected our president with. His vaccine may be different. Maybe, the one they injected him with is not the one they will inject me with for me to become paralyzed. That is why I am scared of taking the vaccine."* (FGD, males above 30 years, Municipality A)

*"My concern is, I am not sick so, why should I take the vaccine and here lies the case, they said after vaccination, I need to wear the nose mask, still sanitize and practice social distancing. So why should I vaccinate? I have been following all these protocols and have not acquired the disease. Why then should I get vaccinated and experience the reactions it comes with?"* (Opinion leader, Municipality A)

Failure to have been engaged about the COVID-19 vaccine caused majority of the participants to reject the vaccine even if that would mean death. Most participants reported that they preferred continuing the protocols and stated that they would not even allow their relatives to take the vaccine.

"*They first have to be educated. Honestly, if you should bring that vaccine tomorrow, everybody will run away.*" (Traditionalist, Municipality A)

"*Even though the coronavirus also leads to death, it should rather kill us when we get infected. It is better than dying from the vaccine.*" (FGD, females above 30 years, Municipality B)

"*I am scared that if I take the injection, I will die and leave my children.*" (FGD, females less than 30 years, Municipality A)

### 3.6 Recommendations from the study participants on how they would like to be engaged concerning misinformation and uptake of the COVID-19 vaccination

Study participants reported that they were ready to take the COVID-19 vaccine once they were properly educated. They stated that this education should be channeled through the community elders and health workers in their communities. They would like to be engaged about the vaccine. A Few participants also recommended that Africans also manufacture their own vaccines and that will motivate them to take the vaccine.

"*For that one, it is necessary that the health directorate officials should come and give us more education. They should come and give detailed education about the vaccine and that is the only way it will be accepted. But if there is no education like how they have been telling us to wash our hands and others, nobody will accept it. More education is necessary". (*Assembly member, Municipal B)

"*Africa should also set up a company that can manufacture vaccines for its members. Madagascar came out with a herbal cure for COVID-19 but the World Health Organization rejected it because it is from Africa. The God who created us, is He not on the side of Africans? Or the brain or knowledge given to the Whites, wasn't it given to Africans? Does it mean that we can only rely on drugs from the Whites and we cannot manufacture any for them to use? These dis-eases like COVID-19, Ebola, and the rest were engineered in the lab by the Whites. They also created COVID-19 to kill us. Why can't African leaders think and come out with their own solutions to this disease? Because it appears it is the same Whites manufacturing the cure for the disease again. If Africans get the cure or the vaccine, we will take it, if not we will not accept it*." (FGD, Male Migrants, Municipality B)

They further added that they would like to be educated on the vaccine and only then would they be willing to agree to be vaccinated. Persons they wanted the government institutions to involve in the engagement process were their chiefs, the community elders, the assemblymen.

"*Sincerely speaking, if the government comes to work with the community to settle on level grounds, then we would change our mindset. However, if they don't come to work with us to settle on level grounds, nothing would change.*" (FGD, males 30 years and above, Municipality B)

In the same vein, all participants reported they would like to be informed about the vaccine. Most study participants stated that they were not engaged in planning of the COVID-19 vaccine rollout but were engaged in activities to fight COVID-19 itself. Regarding the COVID-19 vaccine, few participants reported to have only been informed about it through the radio. However, they reported neither to have been involved in planning, nor been consulted, collaborated with, or empowered to take the vaccine.

"*They also spoke about it on the radio. The District Director of Health Services went on the radio and spoke about it. And so, in our various homes, we listened to the information given to us.*" (Assemblyman, Municipality A)

"*Not yet! They did not take any steps and came to individuals like that yet. You are the first person to ask me concerning these plans…Planning assiduously has not surfaced that I know of…*" (Pastor, Municipality A)

## 4. Discussion

This case study approach used in-depth interviews and focus group discussion to explore fake news and misinformation and how they influenced the decision to vaccinate in two municipalities in Ghana and how community engagement contributed to willingness to vaccinate. The study found that study participants had many sources of information, however, they were predominantly exposed to vaccine-related misinformation through social media, friends, and local media outlets, most of these sources provided fake news, which misinformed them. The extensive dissemination of inaccurate information, primarily through social media and community channels, influenced community members' decisions regarding vaccination. Such misinformation ranged from distortions about the vaccine's intent to outright conspiracy theories concerning its safety and efficacy. This finding corroborates the literature that suggests that social media is a dual-edged sword that offers access to both accurate and misleading information [30–33] The United Nations has noted that this dual exposure complicates the community's ability to distinguish credible information from falsehoods, thereby fostering vaccine hesitancy [34]. According to Skafle et al. [33], misinformation about COVID-19 vaccines spread on social media platforms includes medical misinformation, vaccine development, and conspiracies, with Twitter (now X) being the most studied platform.

The prevalent myths encountered in this study, such as vaccines causing infertility or being a conduit for colonial control, reflect a deep-seated mistrust in external health interventions, a sentiment echoed in the literature as well [35–37] In addition, a critical aspect of the misinformation was the belief that vaccines distributed in Africa were different from those in Europe, intensifying fear and resistance. A similar finding was reported by Gunawardhana et al. [38] that 85% of participants believed the COVID-19 vaccines available in Africa were less effective than those available in Europe. This belief undermines public health efforts and highlights the necessity for transparent communication from health authorities to dispel such myths and reinforce the global standard and safety of the vaccines administered. Moreover, inaccurate information concerning COVID-19 vaccination is more likely to be believed by some conservative religious groups in the US, according to recent studies [39]. Sharing fake news is a real challenge as people cannot distinguish between what is true and what is false which deviates them from searching for information from relevant professionals [40]. Furthermore, considering the powerful influence of religious leaders and their institutions, members who trust these religious leaders are more likely to believe in COVID-19 myths and false information.

Our findings also underscore a significant gender and socio-economic disparity in susceptibility to fake news, with male participants, particularly those with lower educational and economic backgrounds, more prone to accepting and spreading misinformation. This observation aligns with Almenar et al.'s [41] study, which found that men are more likely to receive fake news. Another study highlighted that men were generally more susceptible to gender biases in fake news, indicating a higher prevalence of fake news consumption among male individuals [42]. The challenge here extends beyond mere exposure to misinformation; it involves addressing the foundational reasons that make these demographics more vulnerable, such as lack of access to reliable information and critical engagement with the content they receive.

Despite efforts by the Ghanaian government and health services to combat misinformation through public vaccination campaigns and community engagement, it is evident in our study that these measures were met with scepticism and limited success. This scepticism was partly due to distrust in governmental transparency and the integrity of public health initiatives [43]. The study's results align with findings from De Freitas Silva's [44] study, emphasising that trust in government and health institutions plays a crucial role in public receptiveness to vaccination campaigns. Furthermore, this distrust

had a spillover effect on routine childhood immunizations, which saw a decline as parents conflated their fears about the COVID-19 vaccine with general vaccine mistrust. This trend is alarming and highlights a broader public health crisis where misinformation about a single vaccine affects broader perceptions and acceptance of other essential vaccines.

The distrust of COVID-19 vaccines resulted in community members distrusting health workers because they believed that the health workers were hiding behind the child vaccinations to administer COVID-19 vaccines to their children. Consequently, parents were unwilling to vaccinate their children against the six childhood killer diseases, because they wrongly believed that the children were being offered COVID-19 Vaccines. This development has historical bearing as the region has consistently recorded low rates of childhood vaccination in the country [45]. Also, the WHO reported that COVID-19 lockdowns and fears of its transmission at health facilities contributed to a drop in routine immunization rates between 2019 and 2020, for instance, coverage of the third dose of pertussis-containing vaccine (DTP3), crucial for protecting against diseases like diphtheria and tetanus, declined from 99% to 97%, leaving over 32,000 Ghanaian children vulnerable [30]. Currently, the Ghanaian government with support from the World Health Organization (WHO) and other partners has intensified efforts to strengthen the vaccination program to restore immunization rates by reaching the children who missed vaccinations and to return to pre-pandemic levels [30].

Further, we observed that the pervasive nature of misinformation significantly heightened vaccine hesitancy. Participants expressed fear of severe adverse reactions and doubted the vaccine's efficacy, with many preferring to rely on natural immunity or traditional remedies. This hesitancy is underpinned by a lack of accurate knowledge and an overreliance on anecdotal and often misleading information from non-scientific sources. The gravity of this issue is highlighted in studies by Joseph et al. [46] and Wilson and Wiysonge [47] who noted that misinformation could drastically reduce vaccination rates and increase public health risks. Similarly, other studies have observed that misinformation has led to increased vaccine hesitancy and reduced adherence to public health measures, posing a threat to global health efforts [31,33].

Our study further revealed participants' clear desire for more effective and inclusive strategies to combat misinformation surrounding COVID-19 vaccinations. They recommended that engagement efforts should be spearheaded by trusted community figures such as chiefs, elders, and health workers, who are perceived as more credible and approachable. These figures should be well-equipped with accurate and relevant information to address specific concerns and debunk prevalent myths. This validates Njoga et al.'s [48] study, which echoed the importance of involving traditional and religious leaders in vaccination campaigns. According to them, these leaders, being trusted community figures, play a crucial role in restoring public trust in vaccines. Besides, Cooper et al. [49] emphasised the need for interventions by trusted community members to address vaccine hesitancy and improve vaccination rates. Additionally, the participants suggested that such educational campaigns should be conducted through familiar and accessible platforms, including local community meetings, radio programmes, and social media channels tailored to the local dialects and cultural contexts. Emphasising the importance of transparency, they proposed that the government and health institutions should openly share information about vaccine sourcing, safety protocols, and efficacy studies to build trust. Moreover, there was a strong recommendation for the establishment of local vaccine production to reduce dependency on foreign vaccines, which they believed would significantly increase trust and acceptance among the community members [50,51]. This approach underscores a holistic strategy that not only disseminates information but also fosters a sense of ownership and trust in health interventions.

## 5. Conclusion and recommendation

The section discusses the conclusion and possible recommendations towards improving vaccine uptake in the Volta Region of Ghana.

### 5.1 Conclusion

The study findings revealed that misinformation and fake news about COVID-19 vaccines were widespread in the study communities, with significant implications for vaccine hesitancy. The sources of misinformation ranged from social media

platforms and radio broadcasts to personal interactions within communities. Distrust in government and healthcare institutions, stemming from inconsistent COVID-19 case reports, delayed vaccine rolled-out, and the historical experience with the Ebola trial controversy contributed to vaccine hesitancy. While government efforts at community engagement were noted, these efforts were often inadequate to counteract the deeply ingrained fears and misconceptions. Misinformation negatively influenced uptake of COVID-19 vaccine and consequently, affected childhood immunization programs reflecting a declining trust in public health interventions.

### 5.2 Recommendations

The CHPS programme would need to strengthen community engagement to promote community ownership of education activities. Trusted community leaders such as traditional leaders, religious leaders, and community health workers should be encouraged to carry out education in their communities to counter misinformation.

The health system should develop and disseminate educational materials in local languages through community radio stations, WhatsApp platforms, and songs, leveraging familiar and accessible communication channels.

The government should provide timely, transparent, and consistent information on vaccine procurement and distribution to promote trust. Additionally, the government through the Ghana Health Service should institute avenues for communities to be able to access information on vaccines in order to build public trust. In addition, the Ghana Health Service will need to establish a rapid response system to identify and debunk circulating of fake news through fact-checking platforms and embark on regular community briefings.

The Ghana Health Service needs to commit more resources to training healthcare workers on effective communication strategies to address vaccine hesitancy at the CHPS level.

Another strategy that can contribute to building trust in the medium to long term in the sub-Saharan African region is the sustainable release of funds by governments in the region to support local vaccine research and production. Such an initiative would help to build confidence in health interventions and address concerns about external influence.

### 5.3 Study limitation

In this study as in qualitative studies, a few participants were purposively sampled to participate in the study and for that matter, it cannot be generalized. Nevertheless, our literature suggests that the findings compare with previous studies carried out elsewhere.

### 5.4 Disclosure

COVID-19-related protocols were strictly observed throughout the study. Data collectors used face masks and sanitizers in all the activities. They always observed social distancing. Each study participant was provided with a face mask, and their hands were sanitized before and after the consenting process.

### Acknowledgments

The authors thank the officials of the two municipalities, the participating District Health Management Teams, health facilities, and staff for their support and cooperation. The authors are grateful to community leaders and community members for their cooperation and support.

### Author contributions

**Conceptualization:** Matilda Aberese-Ako, Evelyn Ansah.

**Data curation:** Matilda Aberese-Ako.

**Formal analysis:** Mawulom Kuatewo, Wisdom Ebelin, Lebene Kpodo, Atsu Godsway Kpordorlor, Matilda Aberese-Ako.

**Funding acquisition:** Matilda Aberese-Ako.

**Investigation:** Matilda Aberese-Ako, Evelyn Ansah.

**Methodology:** Matilda Aberese-Ako, Evelyn Ansah.

**Project administration:** Matilda Aberese-Ako, Evelyn Ansah.

**Resources:** Matilda Aberese-Ako.

**Software:** Matilda Aberese-Ako.

**Supervision:** Matilda Aberese-Ako, Evelyn Ansah.

**Validation:** Matilda Aberese-Ako, Evelyn Ansah.

**Visualization:** Matilda Aberese-Ako.

**Writing – original draft:** Mawulom Kuatewo, Wisdom Ebelin, Phidelia Theresa Doegah, Lebene Kpodo, Atsu Godsway Kpordorlor, Samuel Lissah, Senanu Djokoto, Matilda Aberese-Ako.

**Writing – review & editing:** Mawulom Kuatewo, Wisdom Ebelin, Phidelia Theresa Doegah, Lebene Kpodo, Atsu Godsway Kpordorlor, Samuel Lissah, Senanu Djokoto, Matilda Aberese-Ako.

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
