## [Decision Letter · Decision Letter 0]

Dear Dr. Ebelin,

Thank you for submitting your manuscript to PLOS ONE. After careful consideration, we feel that it has merit but does not fully meet PLOS ONE’s publication criteria as it currently stands. Therefore, we invite you to submit a revised version of the manuscript that addresses the points raised during the review process.

We look forward to receiving your revised manuscript.

Kind regards,

Innocent B. Mboya, Ph.D.

Academic Editor

PLOS ONE

**Journal Requirements:**

1. When submitting your revision, we need you to address these additional requirements. Please ensure that your manuscript meets PLOS ONE's style requirements, including those for file naming. The PLOS ONE style templates can be found at https://journals.plos.org/plosone/s/file?id=wjVg/PLOSOne_formatting_sample_main_body.pdf and https://journals.plos.org/plosone/s/file?id=ba62/PLOSOne_formatting_sample_title_authors_affiliations.pdf 2. Thank you for stating the following financial disclosure: Initials of the authors who received each award: MAGrant numbers awarded to each author: Royal Society of Tropical Medicine and Hygiene Early Careergrants programmeThe full name of each funder: Royal Society of Tropical Medicine and HygieneURL of each funder website: https://www.rstmh.org/rstmh-early-career-grants-programme-%E2%80%93-guidance-2024   Please state what role the funders took in the study.  If the funders had no role, please state: "The funders had no role in study design, data collection and analysis, decision to publish, or preparation of the manuscript." If this statement is not correct you must amend it as needed. Please include this amended Role of Funder statement in your cover letter; we will change the online submission form on your behalf. 3. In this instance it seems there may be acceptable restrictions in place that prevent the public sharing of your minimal data. However, in line with our goal of ensuring long-term data availability to all interested researchers, PLOS’ Data Policy states that authors cannot be the sole named individuals responsible for ensuring data access (http://journals.plos.org/plosone/s/data-availability#loc-acceptable-data-sharing-methods). Data requests to a non-author institutional point of contact, such as a data access or ethics committee, helps guarantee long term stability and availability of data. Providing interested researchers with a durable point of contact ensures data will be accessible even if an author changes email addresses, institutions, or becomes unavailable to answer requests. Before we proceed with your manuscript, please also provide non-author contact information (phone/email/hyperlink) for a data access committee, ethics committee, or other institutional body to which data requests may be sent. If no institutional body is available to respond to requests for your minimal data, please consider if there any institutional representatives who did not collaborate in the study, and are not listed as authors on the manuscript, who would be able to hold the data and respond to external requests for data access? If so, please provide their contact information (i.e., email address). Please also provide details on how you will ensure persistent or long-term data storage and availability. 4. PLOS requires an ORCID iD for the corresponding author in Editorial Manager on papers submitted after December 6th, 2016. Please ensure that you have an ORCID iD and that it is validated in Editorial Manager. To do this, go to ‘Update my Information’ (in the upper left-hand corner of the main menu), and click on the Fetch/Validate link next to the ORCID field. This will take you to the ORCID site and allow you to create a new iD or authenticate a pre-existing iD in Editorial Manager. 5. Your ethics statement should only appear in the Methods section of your manuscript. If your ethics statement is written in any section besides the Methods, please delete it from any other section.

Reviewers' comments:

Reviewer's Responses to Questions

**Comments to the Author**

1. Is the manuscript technically sound, and do the data support the conclusions?

Reviewer #1: Partly

Reviewer #2: No

Reviewer #3: Yes

2. Has the statistical analysis been performed appropriately and rigorously?

Reviewer #1: Yes

Reviewer #2: No

Reviewer #3: N/A

3. Have the authors made all data underlying the findings in their manuscript fully available?

Reviewer #1: No

Reviewer #2: No

Reviewer #3: Yes

4. Is the manuscript presented in an intelligible fashion and written in standard English?

Reviewer #1: Yes

Reviewer #2: No

Reviewer #3: Yes

**Reviewer #1: ** The title stresses on childhood vaccination. I am not sure why this is the case. The later part after the quotation marks in the title would suffice.

Again in the discussion, the details of impact of childhood vaccination were discussed in much details and that was not reflected in the results that much.

**Reviewer #2: ** This study is unsuitable for publication in this journal because it is not scientific. Whether this is a fake disease or not should have been discussed in 2020 and 2021. It is better published in a newspaper than in a scientific journal.

**Reviewer #3: ** ABSTRACT

Introduction: Emphatic statement like… According to the World Health Organisation should be reconsidered since there are no in-text citations

Method: There is nothing like qualitative case study. The authors should consider the the use of the term appropriately as case study. In-depth interviews is a procedure and NOT A TOOL. THEREFORE, the use of in-depth interviews in that context of the study is appropriate in the method section of the abstract

The use of community members should be specified since the data were collected from specified participants in the community

Conclusion: This appears more of recommendations and not conclusion. Again, if they are recommendations, they are generally sweeping and should be specified

METHOD

Study design

It will be appropriate to discuss only the study approach here. And discuss the data collection procedure subsequently together with the role of each member. Fusing all together with a heading as study design in my view is problematic.

STUDY SETTING

Is Volta region considered as Southern Ghana? Again, this needs to be justified properly.

2.4. Training of data collectors and quality control

In line 199, the use of validity and reliability is NOT correct SINCE qualitative research cannot establish that in the data collection tool. The authors need to actually explain the rigor employed in the study.

RESULT

This section is detailed enough

However, under line 523, the use of the heading Impact of limited engagement in addressing misinformation is problematic since qualitative study cannot establish impact. Try to rephrase it.

5.0 Recommendation and conclusion

Conclusion should rather come before recommendations

**Do you want your identity to be public for this peer review?** For information about this choice, including consent withdrawal, please see our Privacy Policy

Reviewer #1: No

Reviewer #2: No

Reviewer #3: No

---

## [Author Response · Author response to Decision Letter 1]

19 May 2025

Response to reviewers’ comments

Reviewer #1

Thank you for your careful review, which would go a long way towards improving the quality of the paper. Kindly find below the responses to the issues raised.

Reviewer’s Comment: The title stresses on childhood vaccination. I am not sure why this is the case. The later part after the quotation marks in the title would suffice.

Response: The title has been revised accordingly, kindly see lines 1 and 3

Reviewer’s Comment: Again in the discussion, the details of impact of childhood vaccination were discussed in much details and that was not reflected in the results that much.

Response: The discussion point on childhood vaccination in this current study is based on the results that parents lost trust in health workers as they were afraid that due to their refusal to vaccinate against covid-19, health workers were using the childhood vaccination as an excuse to give their children covid-19 vaccines. Kindly refer to lines 352 to 364.

Reviewer # 2

Thank you for your comments. Kindly find below responses to your comments.

Reviewer’s Comment: This study is unsuitable for publication in this journal because it is not scientific. Whether this is a fake disease or not should have been discussed in 2020 and 2021. It is better published in a newspaper than in a scientific journal.

Response: This study is scientific and suitable for publication in this esteemed journal, because it has utilized systematic and rigorous methods to explore and understand a phenomena as recommended in qualitative studies (Forero et al., 2018, Sale and Thielke, 2018). Importantly, it contributes to health care policy and decision making. Qualitative research focuses on understanding the meanings and experiences of individuals using data collection methods such as in-depth interviews, participant observations, and focus group discussions. These methods generate textual data, which require the use of qualitative data analysis methods such as NVivo. A qualitative software to support coding and analysis. Consequently, this study does not require statistical analysis to make it scientific and relevant to health policy. Additionally, this study adheres to the Consolidated Criteria for Reporting Qualitative Research (COREQ) Checklist (Tong et al., 2007), a recognized scientific and systematic reporting guideline for qualitative research, ensuring methodological rigor and credibility.

COVID-19 was and remains a relevant global public health problem, with many countries still grappling with its effects and seeking effective policy interventions. Ghana has integrated COVID-19 vaccination into routine public health care and thus, it is important that studies of this nature are conducted to inform public health policy. Additionally, the lessons from this study are relevant to other contexts and any other pandemic in the future.

Reviewer #3

Thank you for your careful review, which would go a long way towards improving the quality of the paper. Kindly find below the responses to the issues raised.

ABSTRACT

Reviewer’s Comment: Introduction: Emphatic statement like… According to the World Health Organisation should be reconsidered since there are no in-text citations

Response: The statement 'According to the World Health Organisation' has been removed to align with the absence of an in-text citation. Kindly refer to lines 28 and 29

METHOD:

Reviewer’s Comment: There is nothing like qualitative case study. The authors should consider the use of the term appropriately as case study.

Response: Thank you for the correction. Qualitative has been removed to reflect the design, kindly refer to lines 34, 128 and 622

Reviewer’s Comment: In-depth interviews is a procedure and NOT A TOOL. THEREFORE, the use of in-depth interviews in that context of the study is appropriate in the method section of the abstract

Response: Well noted with thanks

Reviewer’s Comment: The use of community members should be specified since the data were collected from specified participants in the community

Response: Community groups specified, kindly see lines 38

Reviewer’s Comment: Conclusion: This appears more of recommendations and not conclusion. Again, if they are recommendations, they are generally sweeping and should be specified

Response: It is a conclusion. It has been duly revised. Kindly refer to lines 719 - 756

METHOD

Reviewer’s Comment: Study design

It will be appropriate to discuss only the study approach here. And discuss the data collection procedure subsequently together with the role of each member. Fusing all together with a heading as study design in my view is problematic.

Response: The study approach and data collection procedure have been separated accordingly. Kindly refer to lines 127 – 131 concerning the research approach and 185 - 200 for data collection procedure.

STUDY SETTING

Reviewer’s Comment: Is Volta region considered as Southern Ghana? Again, this needs to be justified properly.

Response: Yes, the Volta Region is considered part of Southern Ghana, specifically the southeastern part (Ghana Birth and Deaths Registry, 2025). Geographically, it extends from the Gulf of Guinea in the south, where it has coastal districts, to the northern areas bordering the recently created Oti Region (Nyatuame et al., 2014).

Reviewer’s Comment: 2.4. Training of data collectors and quality control

In line 199, the use of validity and reliability is NOT correct SINCE qualitative research cannot establish that in the data collection tool. The authors need to actually explain the rigor employed in the study.

Response: Authors have explained how rigor was maintained. Kindly refer to lines 212 to 219.

RESULT

Reviewer’s Comment: This section is detailed enough

However, under line 523, the use of the heading Impact of limited engagement in addressing misinformation is problematic since qualitative study cannot establish impact. Try to rephrase it.

Response: The heading has been changed to “Consequences of limited engagement in addressing misinformation on decision to vaccinate”. Kindly refer to line 524

Reviewer’s Comment: 5.0 Recommendation and conclusion

Conclusion should rather come before recommendations

Response: We have revised the section by placing the conclusion section before the recommendations. Kindly refer to lines 722-734 for the conclusion.

Please refer to lines 735– 756 for the recommendations

---

## [Editor Report · Decision Letter 1]

“People will not even bring out their children to be immunised, because of the corona vaccine”: fake news, misinformation, vaccine hesitancy and the role of community engagement in COVID-19 vaccine acceptance in Southern Ghana

PONE-D-24-57065R1

Dear Dr. Ebelin,

We’re pleased to inform you that your manuscript has been judged scientifically suitable for publication and will be formally accepted for publication once it meets all outstanding technical requirements.

Kind regards,

Innocent B. Mboya, Ph.D.

Academic Editor

PLOS ONE

Additional Editor Comments (optional):

Dear Authors,

Thank you for sufficiently addressing the reviewer comments. Please consider removing the preambles before section 2.0 (Methods) and 5.0 (Conclusion and recommendations). Also, I am not sure whether the journal allows section numbers, please check. In addition, Consider removing the section heading "Conclusion and Recommendation" and keep the subsections as they are.

Best wishes.
---

## [Editor Report · Acceptance letter]

PONE-D-24-57065R1

PLOS ONE

Dear Dr. Ebelin,

I'm pleased to inform you that your manuscript has been deemed suitable for publication in PLOS ONE. Congratulations! Your manuscript is now being handed over to our production team.

Kind regards,

on behalf of

Dr. Innocent B. Mboya

Academic Editor

PLOS ONE